# Seeing the Unseen of the Combination of Two Natural Resins, Frankincense and Myrrh: Changes in Chemical Constituents and Pharmacological Activities

**DOI:** 10.3390/molecules24173076

**Published:** 2019-08-24

**Authors:** Bo Cao, Xi-Chuan Wei, Xiao-Rong Xu, Hai-Zhu Zhang, Chuan-Hong Luo, Bi Feng, Run-Chun Xu, Sheng-Yu Zhao, Xiao-Juan Du, Li Han, Ding-Kun Zhang

**Affiliations:** 1Pharmacy College, Chengdu University of Traditional Chinese Medicine; Key Laboratory Breeding Base of Systematic Research and Utilization on Chinese Material Medical Resources Co-founded by Sichuan Province and Ministry of Science and Technology, Chengdu 611137, China; 2Department of Pharmacy and Chemistry, Dali University, Dali 671000, China; 3Chengdu Yong Kang Pharmaceutical Co., Ltd., Chengdu 611130, China

**Keywords:** frankincense, myrrh, combination, phytochemistry, pharmacological activity, synergy, terpenoids, anti-inflammatory, anticancer

## Abstract

For the treatment of diseases, especially chronic diseases, traditional natural drugs have more effective therapeutic advantages because of their multi-target and multi-channel characteristics. Among many traditional natural medicines, resins frankincense and myrrh have been proven to be effective in the treatment of inflammation and cancer. In the West, frankincense and myrrh have been used as incense in religious and cultural ceremonies since ancient times; in traditional Chinese and Ayurvedic medicine, they are used mainly for the treatment of chronic diseases. The main chemical constituents of frankincense and myrrh are terpenoids and essential oils. Their common pharmacological effects are anti-inflammatory and anticancer. More interestingly, in traditional Chinese medicine, frankincense and myrrh have been combined as drug pairs in the same prescription for thousands of years, and their combination has a better therapeutic effect on diseases than a single drug. After the combination of frankincense and myrrh forms a blend, a series of changes take place in their chemical composition, such as the increase or decrease of the main active ingredients, the disappearance of native chemical components, and the emergence of new chemical components. At the same time, the pharmacological effects of the combination seem magically powerful, such as synergistic anti-inflammation, synergistic anticancer, synergistic analgesic, synergistic antibacterial, synergistic blood-activation, and so on. In this review, we summarize the latest research on the main chemical constituents and pharmacological activities of these two natural resins, along with chemical and pharmacological studies on the combination of the two.

## 1. Introduction

In today’s world, cancer and other chronic diseases seriously threaten human health, resulting in an increasing mortality rate year-by-year. Although modern medicine has made great progress in treating these diseases, most FDA-approved single-target drug preparations have some defects in the treatment of complex chronic diseases. Some even have serious side effects [1,2,3]. In traditional medical systems, it is very important to help the body produce self-healing ability in the treatment of chronic diseases. Natural herbal medicines usually have the historical experience of traditional medication and the characteristics of multi-component, multi-link, and multi-target, which make them have potential advantages in the treatment of chronic diseases with relatively low side effects [4,5,6]. Natural medicines widely used in traditional Chinese medicine (TCM) and Ayurvedic medicine for the treatment of chronic diseases include frankincense and myrrh [7,8].

Frankincense and myrrh are two olive plants from different species and genera. Frankincense is a hard, gelatinous resin exuded from the trunk incisions of the frankincense tree, *Boswellia carterii* Birdw. or other species in the genus *Boswellia* of the family Burseraceae, mainly from Somalia, Ethiopia, and India [9]. Myrrh is an oily, gelatinous substance exuded from the bark of *Commiphora myrrha* Engl. or various other species of *Commiphora*, and can be classified as natural myrrh or colloid myrrh. Myrrh exists mainly in tropical and subtropical areas, such as Somalia, Ethiopia, and the southern Arabian Peninsula [10]. In the West, frankincense and myrrh are often used as incense for religious occasions, while in TCM and Indian Ayurvedic medicine, they are used as natural treatments for chronic diseases [11,12,13,14]. 

China is the world’s largest market for frankincense and myrrh, where they are largely consumed as medicinal treatments. In TCM, frankincense and myrrh are both traditional remedies for promoting blood circulation and removing blood stasis. They often appear in the same prescription in the form of drug pairs. They usually have stronger synergistic effects than single medicines. They are mainly used to treat blood stasis, inflammation, swelling, and pain [7]. Prescription drugs containing frankincense and myrrh in combination have definite curative effects on many chronic diseases, and have been clinically proven. For example, Xihuang Pill, a famous anticancer prescription known as the “first drug” of TCM, can effectively treat breast cancer, gastric cancer, and liver cancer, etc.; Xiaojin Pills, the preferred Chinese patented medicine for the treatment of breast hyperplasia, can significantly improve breast diseases; Huoluo Xiaoling Dan also has a good therapeutic effect on chronic diseases, such as arthritis [15,16,17]. In addition, the combination of frankincense and myrrh has been documented in the ancient Egyptian prescription collection of Papyrus Ebers as a treatment for wounds and skin ulcers [18]. And frankincense and myrrh in combination have superior healing properties on oral ulcer compared to triamcinolone acetonide [19].

In the past, most of the studies on frankincense and myrrh focused on single-flavor medicines, and all aspects of research were relatively in-depth. However, in recent years, with the rise of network sciences such as systems biology and network pharmacology, drug combination has become a hot topic in modern drug research and a new trend in the development of modern drugs [20,21]. The synergistic effects of the combination formed by these two natural resins have been confirmed and have attracted worldwide attention. More and more researchers are actively exploring the substance basis and mechanism of the combination of frankincense and myrrh. Founded on this, this paper systematically reviews the chemical and pharmacological studies of frankincense, myrrh, and frankincense–myrrh compound in an effort to unravel the mysteries of the synergistic effects of the compound from the point of view of chemistry and pharmacology, in order to provide reference for the further study of frankincense, myrrh, and frankincense–myrrh compound, and to provide a relevant basis for their use in the clinical treatment of diseases.

## 2. Frankincense (Olibanum)

### 2.1. Main Chemical Constituents of Frankincense

Modern studies have shown that the main chemical constituents isolated from frankincense are pentacyclic triterpenoids (**1**–**25**), tetracyclic triterpenoids (**26**–**37**), macrocyclic diterpenoids (**38**–**51**), and a variety of essential oils (**52**–**63**). Pentacyclic triterpenoids are the most characteristic and deeply-studied components in frankincense. According to their structures, they can be divided into ursolidine (**1**–**16**), oleanolic (**17**–**20**), and lupinane (**21**–**25**). Their representative compounds are β-boswellic acid (**1**), acetyl-β-boswellic acid (**2**), 11-keto-β-boswellic acid (KBA, **3**), 3-acetyl-11-keto-β-boswellic acid (AKBA, **4**), α-boswellic acid (**17**), and acetyl-α-boswellic acid (**18**), which have been identified as the main components and are considered to be the biomarkers of frankincense [8]. The main chemical constituents and plant sources of frankincense are shown in Table 1, and the structural patterns of some chemical constituents are shown in Figure 1.

### 2.2. Pharmacological Activity of Frankincense

In recent years, the pharmacological research on frankincense has focused mainly on the anti-inflammation and anticancer effects of extracts and chemical components, along with other pharmacological effects, including anti-ulcer, memory improvement, and anti-oxidation. The pharmacological effects and mechanisms of frankincense are shown in Table 2.

#### 2.2.1. Anti-Inflammatory

Frankincense extract exerts anti-inflammatory activity against a variety of acute and chronic inflammatory models. The initial study showed that KBA (**3**) in frankincense had a unique 5-lipoxygenase (5-LO), double inhibitory effect on human leukocyte elastase (HLE), and the inhibitory effect on 5-LO was specific, having no effect on the activities of cycloenzyme or 12-lipoxygenase [50]. Further studies showed that the anti-inflammatory effect of frankincense was related to its inhibition of 5-lipoperoxidase (5-LOX) catalyzed biosynthesis [51]. Through the regulation of inflammatory cytokines and protein kinase pathway by boswellic acids (Bas), Liang et al. found that the combination of AKBA (**3**) and arsenic trioxide could inhibit the secretion of matrix metalloproteinase-1(MMP-1), MMP-2, MMP-9, tumor necrosis factor-α (TNF-α), and interleukin-1beta(IL-β) [52]. Rats with arthritis induced by adjuvant were treated with frankincense extract at a dose of 0.90 g⋅kg^−1^ per day for 10 days, and the results showed that frankincense extract had significant anti-inflammatory effect and that its mechanism might be related to the inhibition of pro-inflammatory cytokines [53]. 

Recently, Henkel et al. selected human antimicrobial peptide LL-37 from human neutrophils as the target of Bas by using KBA (**3**) as bait in an unbiased target fishing method, which indicated that the anti-inflammatory mechanism of Bas was related to the inhibition of antimicrobial peptide LL-37 by Bas, and that Bas could be used as a potential drug to interfere with LL-37 and related diseases [54]. Li et al. used frankincense oil extract (FOE) and its active components α-pinene (**52**), linalool (**61**), and 1-octanol (**63**) to treat ear edema model and formalin inflamed posterior claw model mice according to 0.15 mL⋅cm ^−2^, and the results showed that they could significantly reduce swelling and pain and had significant local anti-inflammatory and analgesic effects. The possible mechanism is to inhibit inflammatory infiltration induced by nociceptive stimulation and overexpression of cyclooxygenase-2 (COX-2) [55]. 

#### 2.2.2. Anticancer

The active anticancer components of frankincense are mainly essential oil, macrocyclic diterpenes, and pentacyclotriterpenes, among which the anticancer activity of Bas in pentacyclotriterpenes is most often reported.

The essential oil of frankincense has specific cytotoxicity to tumor cells and can distinguish normal cells from bladder cancer cells. At a certain concentration, it can specifically block the growth cycle of bladder cancer cell line J82, inhibit cell growth, and induce apoptosis. [56]. The essential oil of frankincense can inhibit the proliferation and induce apoptosis of human hepatocellular carcinoma cell line SMMC-7721. The mechanism may be to induce apoptosis of SMMC-7721 cells by up-regulating the expression ratio of bax/bcl-2 in mitochondria, and the apoptosis induced by frankincense is cell cycle-dependent [57]. Recent studies have found that frankincense essential oil can effectively inhibit the growth of breast cancer (BC) cells and induce apoptosis of BC cells by regulating AMPK/mTOR pathway [58]. 

The 50% inhibitory concentration (IC_50_) values of macrocyclic diterpenes incensole acetate (**45**) and incensole (**44**) were 68.8 μg⋅ml ^−1^ and 39.2 μg⋅mL^−1^, respectively, and incensole acetate also inhibited the growth of human leukemia HL-60 cells with an IC_50_ value of 16.3 ± 3.4 μmol⋅L^−1^. The antitumor effect of macrocyclicditerpenes is similar to that of frankincense, and the amount of diterpenes is high, so it has certain potential for the development of antitumor drugs [44,45]. 

Triterpenoids, especially AKBA (**4**) and KBA (**3**), have significant antitumor activity. Takada et al. found that AKBA (**4**) could promote apoptosis induced by tumor necrosis factor (TNF), weaken the activation of nuclear factor-kappa B (NF-kB) and the expression of NF-kB regulatory gene, and prevent osteoclast formation [59]. AKBA (**4**) has a significant anti-prostate cancer effect, which can inhibit the growth of prostate cancer cells and induce apoptosis of prostate cancer cells. Its mechanism is related to the inhibition of the vascular endothelial growth factor receptor 2 (VEGFR2) signaling pathway and the up-regulation of the death receptor, DR5 pathway [60,61,62]. Park et al. found that AKBA (**4**) inhibition of breast cancer cells was associated with the disappearance of the chemokine receptor 4 (CXCR4) signal RNA and CXCR4 protein [63]. Shen et al. treated RKO, SW48, and SW480 colorectal cancer (CRC) cell lines with AKBA (**4**) and found that their anticancer effect may be partly due to their ability to demethylate and reactivate the silenced tumor inhibitor genes [64]. Zhao et al. reported that KBA (**3**) could inhibit the growth of B16F10 mouse melanoma cells and MV-3 human melanoma cells, induce cell morphological changes, weaken cell motility, and increase the amount of melanin [65]. At the same time, KBA (**3**) could also reduce the malignant degree of BGC-803 and BGC-823 in human gastric cancer cells. KBA (**3**) can induce differentiation of HL-60, U937, and ML-1 in myeloid leukemia monocytes at 24.2 μmol⋅L^−1^, resulting in 90% cell morphological changes, and the positive rate of NBT (fructosamine diagnostic kit) is 80–90% [66]. In addition, the antitumor effect of Bas is partly due to their ability to up-regulate the expression of let-7 and miR-200microRNA families [67].

#### 2.2.3. Other Pharmacological Activities

It has been confirmed that AKBA (**4**) can regulate the activity of human collagenase MMP-1, MMP-2, and MMP-9. Therefore, the direct inhibition of MMP activity and the inhibition of MMP secretion may be one of the mechanisms of frankincense in the treatment of chronic skin ulcers, and reducing the high MMP activity of the wound surface is a new way to treat chronic skin ulcers [68]. Rats with hypothyroidism induced by methimazole were treated with oral frankincense (100 or 500 mg⋅kg^−1^ per day) for 180 days. The results of the Morris water maze test showed that the swimming speed of rats in the frankincense group was significantly improved, suggesting that frankincense could prevent the symptoms of learning and memory loss [69]. Recent studies have shown that Bas can eliminate ROS and inhibit lipid peroxide and DOX-induced hepatotoxic DNA damage. The antioxidant effect of Bas may be due to their regulation of the Nrf2/HO-1 pathway, thus protecting the liver from DOX-induced oxidative damage [70]. 

## 3. Myrrh

### 3.1. Main Chemical Constituents of Myrrh

Modern studies have shown that myrrh is 3–8% essential oil, 25–40% alcohol-soluble resin, and 30–60% water-soluble gum [10]. The chemical constituents of essential oils include monoterpenes (**64**–**69**), sesquiterpenes (**70**–**80**), and small molecular aromatic compounds. Chemical constituents in resins include diterpenoids (**81**–**85**), triterpenoids (**86**–**95**), steroids (**96**–**101**), and lignans (**101**–**107**). The main components of gum are sugar, protein, and ash. The main chemical constituents and plant sources of myrrh are shown in Table 3, and the structural patterns of some chemical constituents are shown in Figure 2.

### 3.2. Pharmacological Activity of Myrrh

Modern studies have shown that the main pharmacological effects of myrrh are anti-inflammatory, anticancer, analgesic, and antibacterial, among which the anticancer effect has been studied the most. The pharmacological action and mechanism of myrrh are shown in Table 4.

#### 3.2.1. Anti-Inflammatory

Myrrh contains many active ingredients with strong anti-inflammatory effects, among which myrrh steroid, guggulsterone (GS), can improve acute pancreatitis. Kim et al. found that intraperitoneal injection of GS 10, 25, or 50 mg⋅kg^−1^ 1 h before the establishment of the model could effectively treat acute pancreatitis by inhibiting the activation of extracellular-regulated protein kinases (ERK) and c-Jun N-terminal kinase (JNK) [98]. Duwiejua et al. found that the anti-inflammatory potency of the triterpenoid manumbinoic acid (**91**) isolated from *Commiphora incisa* resin is in the same order of magnitude as indomethacin and prednisolone, and oral administration of mansumbinoic acid (1.5 × 10^−4^ mol·kg^−1^ per day) could significantly reduce the swelling of joints in mice with induced arthritis, which shows that mansumbinoic acid has the potential to become an anti-inflammatory drug [99]. Bellezza et al. found that pretreatment with 1(10), 4-furanodien-6-one (**78**) from *Commiphora erythraea* could restore the viability and ROS control level of microglia BV-2 cells, reduce NO production, and significantly reduce the levels of pro-inflammatory cytokines IL-6, IL-23, IL-17, TGF-B, and INF-gamma induced by lipopolysaccharide (LPS). At the same time, in vivo studies found that the expression of TNFα and IL-1β in the liver and brain decreased after intraperitoneal injection of 1(10), 4-furanodien-6-one (**78**) (2.4 mg⋅kg^−1^) in a mouse model induced by LPS. In vitro and in vivo results indicate that 1(10), 4-furanodien-6-one (**78**) has anti-inflammatory activity, inhibits NF-κB signaling, and attenuates LPS-induced neuro-inflammation [100].

#### 3.2.2. Anticancer

Modern pharmacological and clinical studies have shown that elemene in myrrh has a good anticancer effect. β-elemene (**70**) has been used as an anticancer drug in the treatment of various cancers, including glioblastoma, and the anti-proliferation effect of β-elemene on glioblastoma is realized by activating p38MAPK [101]. A furose-type sesquiterpene rel-1S,2S-epoxy-4R-furanogermacr-10(15)-en-6-one (**76**) in myrrh has weak cytotoxicity against breast cancer cell line MCF-7 (IC_50_ = 40 μmol·L^−1^), and seven cyclobolinane triterpenoids in myrrhytrid are moderately cytotoxic to PC3 and DU145 human prostate cancer cells (IC_50_ values ranging from 10.1 to 37.2 microM ^1^) [93]. Yeo et al. found that the sesquiterpene compound β-bisabolene (**80**) in *Commiphora guidotti* has selective cytotoxicity to mouse cells (IC_50_ in normal Eph4: >200 μg⋅mL^−1^, MG1361: 65.49 μg·mL^−1^, 4T1: 48.99 μg·mL^−1^) and breast cancer cells (IC_50_ in normal MCF-10A: 114.3 μg·mL^−1^, MCF-7: 66.91 μg·mL^−1^, MDA-MB-231: 98.39 μg·mL^−1^, SKBR3: 70.62 μg·mL^−1^, and BT474: 74.3 μg·mL^−1^). In vivo studies showed that intraperitoneal injection of 1.12 g⋅kg^−1^ β-bisabolene (**80**) twice a week for two weeks could effectively reduce the growth of 4T1 breast tumors transplanted into model mice (37.5% reduction in end volume). These results indicate that beta-bisabolene could be used as a candidate drug for anti-breast cancer [83]. Myrrh triterpenoids cycloartan-24-ene-1α,2α,3β-triol (**95**) is cytotoxic to human prostate cancer PC-3 cells (IC_50_ = 9.6 μM), which can exert significant apoptotic activity and have potential as new anticancer drugs [102]. Mahmoud et al. found that rats with liver cancer induced by diethylnitrosamine (DEN)/phenobarbital (PB) can be given 125 or 250 mg⋅kg^−1^
*Commiphora molmol* resin extract daily for 14 weeks to prevent early liver cancer. Its mechanism may be related to up-regulation of Nrf2/HO-1 signaling and reduction of inflammation, angiogenesis, and oxidative stress [103].

Recent studies have shown that myrrh steroid GS has potential antitumor activity. It inhibits the regulation of cyclin, inhibits the growth of various tumor cells, and induces apoptosis by down-regulating anti-apoptotic gene products (IAP1, xIAP, Bfl-1/A1, Bcl-2, cFLIP, and survivin) [104]. Shi et al. found that GS (5–100 μmol⋅L^−1^) exerts its anticancer effect by inhibiting cell proliferation and inducing apoptosis of HepG2 cells, and it induces apoptosis by activating the intrinsic mitochondrial pathway [105].

#### 3.2.3. Analgesic Effect

Myrrh has been used as an analgesic since ancient times, and modern studies have found that the sesquiterpenes furanocudesma-1,3-diene (**72**) and curzerene (**74**) in myrrh can act on opioid receptors in the central nervous system and have analgesic activity, which can be blocked by morphine antagonist naloxone [106]. Mehta et al. took *Commiphora mukul* (CM) extract orally at doses of 50, 100, and 200 mg⋅kg^−1^ per day for 2 weeks. They found that CM (50 mg⋅kg^−1^) had a significant effect on alleviating cold abnormal pain; CM (100 mg⋅kg^−1^) and CM (200 mg⋅kg^−1^) can significantly alleviate heat hyperalgesia and cold abnormal pain. These results suggest that CM can alleviate peripheral nerve pain caused by chronic compressive injury of the sciatic nerve and can be used as an alternative drug for the treatment of nerve pain in the future [107]. 

Germano et al. found that MyrLiq, a *Commiphora myrrha* extract containing furanocudesma-1, 3-diene (**72**), lindestrene (**73**), curzerene (**74**), and a high content of total furanodiene had significant analgesic effects. When male volunteers took 400 mg of MyrLiq per day for 20 days, relief was achieved for almost all pain conditions, while for female volunteers, only 200 mg of MyrLiq per day for 20 days was taken, and relief for low back pain and fever-dependent pain was observed [108].

#### 3.2.4. Antibacterial

Kaleab et al. found that the essential oil of *Commiphora tenuis* Vollensen resin has antibacterial activity against *Staphylococcus aureus*, *Proteus mirabilis*, and *Escherichia coli*, and the minimum inhibitory concentration (MIC) is 0.5–1% [73]. Rahman et al. isolated four antimicrobial terpene compounds from *Commiphora mol*, mansumbinone (**89**), 3,4-seco-mansumbinoic acid (**90**), β-elemene (**70**), and T-cadinol (**71**). Among them, 3,4-seco-man-sumbinoic acid (**90**) has the strongest antimicrobial activity. Its antimicrobial activity against *Staphylococcus* is eight times that of Norfloxacin, and it has a weak enhancement of the antimicrobial activity of ciprofloxacin and tetracycline against Salmonella strains SL1344 and L10 [88].

## 4. Frankincense–Myrrh Compound

### 4.1. Chemical Constituents of the Combination

The chemical composition of the combination formed by frankincense and myrrh is different from the chemical composition of the two single-flavor drugs, but it is not a simple addition of the chemical components of the two drugs. The complex physical and chemical changes before and after the combination of frankincense and myrrh make the proportion and content of each component in the compound change to a certain extent. While some original components may be lost, other new components may be produced (Figure 3).

#### 4.1.1. Changes of Terpenoids in the Combination

Terpenoids are one of the main active ingredients of frankincense and myrrh, and their changes significantly affect the pharmacological effects. Chen et al. used the UPLC-Q-TOF-MS/MS to study the dissolution of the chemical components before and after the compatibility of the frankincense and the myrrh, and through OPLS-DA analysis and chemical identification, they found that the contents of the main active ingredients, pentacyclic triterpenoids and tetracyclic triterpenoids (elemonic acid (**118**); 3-hydroxylanosta-8,24-dien-21-oic acid; 2-methoxy-5-acetoxy-fruranogermacr-1(10)-en-6-one; 3-hydroxytirucalla-8,24-dien-21-oic acid; 3-epi-lupenylacetate; and 3-acetoxy-16-hydroxyl-dammar-24-ene), increased significantly after compatibility (1:1), while the dissolution contents of some components of cyclic sesquiterpenoids and macrocyclic diterpenoids(2-acetoxy-furanodiene; 3,17-dihydroxy-3-pregn-5-en-20-one; incensole acetate (**45**); and 15-hydroxyl-mansumbione) decreased [109]. Further, UPLC-TQ/MS was used to quantitatively analyze 11 terpenoids in frankincense after the combination of frankincense and myrrh, and it was found that the contents of 3-acetoxy-tirucall-7,24-dien-21-oic acid in frankincense increased significantly (from 1.103 to 1.916%) after compatibility, and that the contents of β-boswellic acid (**1**), 3α-acetyl-20(29),-lupine-24-oic acid, and 3-acetyl-11-keto-β-boswellic acid (**4**) increased slightly (from 0.919, 0.178, and 1.365% to 1.156, 0.220, and 1.752%, respectively); the contents of 3β-acetoxy-5α-lanosta-8,24-dien-21-oic acid, acetyl 11α-methoxy-β-boswellic acid, and 3α -acetyloxylanosta-8,24-dien-21-oic acid declined significantly (from 0.334, 1.010, and 0.815% to 0.210, 0.534, and 0.622%, respectively), and the contents of α-boswellic acid (**17**) and 3-keto-tirucall-8,24-dien-21-oic acid (**32**) decreased slightly (from 0.553 and 1.071% to 0.505 and 0.971%, respectively) [110]. 

#### 4.1.2. Changes of Essential Oil Components in the Combination

Essential oil is one of the main medicinal ingredients of frankincense and myrrh, and its composition changes have an effect on the efficacy of frankincense and myrrh in combination. By GC-MS analysis for comparison of the components and contents of essential oil before and after the compatibility of frankincense and myrrh, it was found that m-xylene, 1-methoxy-4-(4-methylphenoxy) benzene were added to the essential oil of frankincense and myrrh [111]. 

#### 4.1.3. Potential Bioactive Components of the Combination

In their study of the combined water extract of frankincense and myrrh (CWE) on anti-inflammatory and analgesic activities in vivo, Su et al. analyzed and identified 12 major potentially bioactive components in frankincense and myrrh combined by water extraction with ultra-high performance liquid chromatography-mass spectrometry (UPLC-MS/MS) as follows: Acetyl-11-keto-β-boswellic acid (**3**); 9,11-dehydro-β-boswellic acid (**13**); α-boswellic acid (**17**); 3-keto-tirucall-8,24-dien-21-oic acid (**32**); 2-methoxy-8,12-epoxygermacra-1(10),7,11-trine-6-one (**108**); 7-methoxy-3,6,9-trimethyl-6,6α,7,8,9,9α-hexahydroazuleno[4,5-β]furan-4(5H)-one (**109**); 2R-methoxy-4R-furanogermacr-1(10)E-en-6-one (**110**); 2-acetoxyfuranodiene (**111**); 3,17-dihydroxy-3β-pregn-5-en-20-one (**112**); 1,2,3-trihydroxyurs-12-en-28-oic acid (**113**); 3-keto-tirucall-7,24-dien-21-oic acid (**114**); and 3-hydroxytirucall-8,24-dien-21-oic acid (**115**) [112]. Studies on the anti-inflammatory mechanism of frankincense and myrrh showed that five compounds—3-hydroxylanosta-8,24-dien-21-oic acid (**116**); 2-methoxy-5-acetoxy-fruranogermacr-1(10)-en-6-one (**117**); abietic acid (**84**); elemonic acid (**118**); and acetyl elemolic acid (**119**)—inhibited phosphorylation levels of ERK, JNK, and p38 in peripheral blood mononuclear cells (PBMC) activated by PHA. Especially acetyl elemolic acid (**119**) has a more significant inhibitory effect on pro-inflammatory factors [113]. In addition, Hu et al. identified five analgesic substances in water extract of frankincense–myrrh (WFM) by UHPLC-TQ/MS, and they are β-boswellic acid (**1**); 3-acetyl-11-keto-β-boswellic acid(**4**); 3-keto-tirucall-8,24-dien-21-oic acid (**32**); abietic acid (**84**); and 3α-acetoxy-tirucall-7,24-dien-21-oic acid (**120**) [114]. The chemical structure of some potential bioactive components in the frankincense–myrrh compound is shown in Figure 4.

### 4.2. Pharmacological Activity of the Combination

The abundant clinical experience shows that the combined use of frankincense and myrrh is not a simple superposition of the pharmacological effects of the two drugs, but a synergistic effect that increases the pharmacodynamics, including synergistic anti-inflammatory, synergistic anticancer, synergistic analgesic, synergistic antibacterial, and synergistic blood-activating effects. In addition, the frankincense–myrrh compound also has a significant penetration-promoting effect in vitro. These synergistic pharmacological effects have been confirmed more and more in modern experimental research, and the related mechanisms are also being gradually improved (Figure 5).

#### 4.2.1. Synergistic Anti-inflammatory

Modern study shows that both the frankincense and the myrrh have good therapeutic effects on inflammation, and the combination of the two drugs has even better synergistic anti-inflammatory activity. Studies on the anti-arthritis effects of single and combined extracts of frankincense and myrrh showed that the expression levels of inflammatory cytokines (INF-γ, IL-2, IL-1β, IL-12, TNF-α, PGE2, NO, and MDA) in serum and foot swelling of adjuvant-induced arthritis (AIA) in rats were significantly decreased, and the inhibitory effect of the compound therapy was more obvious after being given the combined extracts of frankincense and myrrh orally (54.28 mg⋅kg^−1^ per day, 90.48 mg⋅kg^−1^ per day), frankincense extracts (33.67 mg⋅kg^−1^ per day, 56.12 mg⋅kg^−1^ per day), and myrrh extracts (46.15 mg⋅kg^−1^ per day, 76.92 mg⋅kg^−1^ per day) for 17 consecutive days. By analyzing the expression of inflammatory cytokines, *c-jun* and *c-fos*, and the metabolic spectrum and signal transduction pathway of phosphorylation level assessment, it was found that the bioactive components of frankincense and myrrh play an anti-inflammatory role by reducing the phosphorylation forms of all three kinds of MAPK (ERK, p38, and JNK) and the down-regulation of downstream genes (*c-jun* and *c-fos*) [113] (Figure 6).

Su et al. found that the combined extract (CWE) had synergistic effects and enhanced anti-inflammatory activity in the treatment compared with individual herbal extracts by using paw edema mice to compare the anti-inflammatory activities of myrrh water extract (MWE), frankincense water extract (FWE), and the combined water extract (CWE). After intragastric administration of MWE 3.9 g⋅kg^−1^, FWE 6.8 g⋅kg^−1^, and CWE 5.2 g⋅kg^−1^, the results showed that MWE and CWE could inhibit formalin-induced paw edema in mice. At the same dosage of intragastric administration, MWE, FWE, and CWE significantly inhibited the development of carrageenan-induced paw edema after 1, 2, and 3 h of administration, and CWE had stronger inhibitory effect on paw edema in mice at 2 and 3 h compared with MWE and FWE, showing enhanced inhibitory effect [112].

#### 4.2.2. Synergistic Anticancer

Frankincense and myrrh have anticancer activities against breast cancer cells, and the combination of frankincense and myrrh has synergistic anticancer effect, which is better at killing breast cancer cells. Through network pharmacological analysis, it was found that the anti-breast cancer process of frankincense and myrrh involved 10 active ingredients such as diayangambin (**107**) and guggulsterone (**96**,**97**), 43 protein targets such as ATR and TP53, and 25 signaling pathways such as cAMP signaling pathway and PI3K-Akt signaling pathway, which could provide a reference for the study of the material basis and mechanism of anti-breast cancer of frankincense–myrrh compound (Figure 7).

Cheng et al. found that the extracts of frankincense and myrrh (0.25–4 mg⋅mL^−1^) had significant inhibitory effects on the proliferation of human breast cancer cell line MCF-7, human breast cancer cell line MDA-MB-231, and lung cancer cell line NCI-H446. For the same cell line, the inhibitory effects of frankincense and myrrh combined extracts were stronger than those of frankincense or myrrh alone [115]. In addition, the frankincense–myrrh compound also has significant anticancer activity against other cancer cells. In vitro anticancer studies showed that the combined extracts of frankincense and myrrh at concentrations of 10 mg⋅mL^−1^ and 20 mg⋅mL^−1^ inhibited the proliferation of human hepatocellular carcinoma cell line SMMC7721 and human promyelocytic leukemia cell line HL-60, and the activity of receptor tyrosine kinase (RTKs) in SMMC7721 and HL-60 cells. In vivo studies found that tumor growth was significantly inhibited in H22 tumor-bearing mice after 10 days of oral administration of the combined extracts of frankincense and myrrh (1.08 g⋅kg^−1^ per day, and 2.16 g⋅kg^−1^ per day), and the activity of RTKs in tumor tissue was also significantly decreased. These results suggest that the combination of frankincense and myrrh may play an anticancer role by inhibiting RTKs activity and will hopefully develop into a new receptor tyrosine protein kinase inhibitor [116].

In a study of the efficacy of the extract of frankincense–myrrh compound in the regulation of antitumor immune response in the hepatocellular carcinoma (HCC) model, it was found that a non-toxic dose of the extract of frankincense–myrrh compound (0.5 mg⋅mL^−1^) could significantly inhibit the activation of the NF-κB and STAT3 signals of the liver cancer cell line HCCLM3 and Hepa1-6 induced by the cytokine (TNF-α or IL-6), and inhibit the activation of the NF-κB and STAT3 signals in the co-culture system with the CD8 ^+^ NKG2D ^+^ cells. In vivo, immunocompromised liver cancer-bearing mice and immunoactive liver cancer-bearing mice were treated with the extract of frankincense–myrrh compound 60 mg⋅kg^−1^ daily until the specified time point. The results showed that the extract of frankincense–myrrh compound could not inhibit the tumor growth in immunodamaged mice, but it could significantly inhibit the tumor growth and life span of immunoactive mice [117].

#### 4.2.3. Synergistic Analgesic

Modern studies have shown that the combined water extract of frankincense and myrrh has a synergistic analgesic effect. For three days, the dysmenorrhea model mice received intragastric administration of myrrh water extract (MWE) 3.9 g⋅kg^−1^ per day, frankincense water extract (FWE) 6.8 g⋅kg^−1^ per day, or the combined water extract (CWE) 5.2 g⋅kg^−1^. The results showed that CWE showed obvious synergistic analgesic effect, which could significantly shorten the amount of writhing and prolong the latency, and MWE could significantly shorten the amount of writhing, while FWE had no significant effect on the amount of writhing [112].

In a study of analgesic mechanisms, Hu et al. observed the effect of a combined water extraction of frankincense and myrrh (WFM) on neuropathic pain sensitization by establishing a chronic constraining injury (CCI) model of the sciatic nerve in mice. The study found that WFM (1.5 g⋅kg^−1^ per day, 7.5 g⋅kg^−1^ per day) had a good analgesic effect on CCI mice after 10 days of intragastric administration, and the expression of Transient receptor potential vanilloid 1 (TRPV1) decreased significantly in real-time PCR, Western imprinting, and immunofluorescence staining, suggesting that WFM plays an analgesic role mainly by regulating the expression and activity of TRPV1 [114].

#### 4.2.4. Synergistic Antibacterial

The antibacterial activity in vitro showed that more than half of the three essential oils of frankincense (*Boswellia rivae, Boswellia neglecta,* and *Boswellia papyrifera*) and two kinds of essential oils of myrrh (*Commiphora guidotti* and *Commiphora myrrha*) had some favorable antibacterial interactions (11.1% synergistic effect and 41.7% additive effect) with no antagonistic effects. In addition, the synergistic effect of the combination of *Boswellia papyrifera* and *Commiphora myrrha* was the most significant (mean ∑FIC was 0.67) in the study of different drug combinations against *Bacillus cereus.* These results further prove the value of the traditional use of the combination of frankincense and myrrh essential oils, which will hopefully become an antimicrobial agent [118].

#### 4.2.5. Synergistic Blood-Activating

Jiang et al. used an in vitro anti-platelet aggregation test induced by the adenosine diphosphate (ADP) and thrombin time (TT) method to observe the anti-platelet aggregation activity of frankincense, myrrh extract, and different combinations (frankincense water extract + myrrh water extract; frankincense essential oil + myrrh essential oil; frankincense water extract + myrrh essential oil; and frankincense essential oil + myrrh water extract) and their effects on thrombin. They used the equivalent line method to evaluate the pharmacodynamic interaction of the two drugs. Their results showed that the extracts of frankincense, myrrh, and their different combinations (2 g·mL^−1^ and 1 g·mL^−1^) could significantly or very significantly inhibit ADP-induced platelet aggregation in rabbits. The two drugs had synergistic effects (interaction coefficient γ < 1), and their activities were stronger than those of the single extracts. At the same time, different extracts from frankincense and myrrh could significantly prolong the plasma coagulation time in rabbits. With the exception of frankincense water extract and myrrh water extract (γ > 1), the other compatibility combinations had synergistic effects (γ < 1) [119].

#### 4.2.6. Penetration-Promoting Effect In Vitro

Zhu et al. studied the effect of the essential oils of frankincense and myrrh on the transdermal properties of Chuanxiong in vitro through improved Franz diffusion cell and isolated mouse skin. The results showed that the combination of frankincense and myrrh essential oils had a certain osmotic promoting effect on ferulic acid, the index component of Chuanxiong, and the combined extraction of essential oils had the strongest osmotic-promoting effect (the transdermal enhancement multiple was 8.28) [120]. Additionally, laser doppler blood flow measurement showed that the frankincense and myrrh essential oil compound could promote the elimination of capillaries from skin epidermis to dermis by increasing skin blood flow. Further studies on the mechanism of penetration showed that the essential oils of frankincense and myrrh may change the conformation of lipids and keratins in the cuticle, increase the fluidity of the lipid bilayer in the cuticle, and change the orderly and dense structure so as to increase the permeability of the skin and decrease the barrier effect [121].

## 5. Conclusion and Prospect

In this review, we summarized two important natural resins for the treatment of chronic diseases in traditional medicine, focusing on modern studies of the chemical and pharmacological effects of frankincense, myrrh, and their combination. In summary, frankincense and myrrh have a wide variety of chemical constituents and a wide range of pharmacological activities. The main bioactive substances of frankincense are Bas whose main pharmacological effects are anti-inflammatory and anticancer, among which AKBA (**4**) and KBA (**3**) have the strongest anticancer activity and are expected to become candidates for anticancer drugs. The main bioactive substances of myrrh are sesquiterpenes and GS (**96**,**97**), and its main pharmacological action is anticancer. When they are paired to form a compound, their chemical composition and pharmacological effects change to some extent. Worthy of note is the acetyl elemolic acid (**119**) in the combination, which has significant anti-inflammatory activity. The pharmacological properties of the combination are mostly synergistic, including synergistic anti-inflammatory activity, synergistic anticancer, synergistic analgesic, synergistic antibacterial, and synergistic blood-activating effects.

Study on the material basis and synergistic mechanism of the frankincense–myrrh combination is still in the initial stage. It was found that the dissolution of sesquiterpenes decreased before and after the combination of frankincense and myrrh, but sesquiterpenes were the strongest active ingredient to inhibit the release of NO from mouse macrophages [109]. This suggests that a series of changes in the chemical composition of frankincense and myrrh may be related to the chemical reaction and physical changes in the merging process, such as solubilization, oxidation, reduction, hydrolysis, etc. [122]. Whether the changes of various components are the material basis of the synergistic effect of frankincense–myrrh compound needs further study. In recent years, the mechanism of action of frankincense and myrrh has been studied mostly by separating chemical constituents and screening the simple active substances. However, this is unsuitable for the characteristics of complex components and multi-target synergism of frankincense and myrrh. Therefore, it is suggested that future studies be based on the chemical and pharmacological studies of frankincense and myrrh, combined with new methods, such as systematic biology and metabolomics, to elucidate the material basis and mechanisms of action of frankincense and myrrh and their combination [123,124].

In brief, frankincense and myrrh, as two traditional natural medicines, have extensive and significant pharmacological effects, and their combination has magically synergistic effects. Therefore, based on the research value and great prospects of the significant efficacy of frankincense–myrrh compound for various chronic diseases, more in vitro, in vivo, and clinical studies are needed to verify the synergistic efficacy and safety of the frankincense–myrrh compound. At the same time, it is necessary to make further use of modern drug research methods to develop and utilize active ingredients rationally, develop them directly into new drugs, or design and synthesize more effective new drugs with active ingredients as leading compounds, thus expanding the clinical application of frankincense and myrrh.

## Figures and Tables

**Figure 1 molecules-24-03076-f001:**
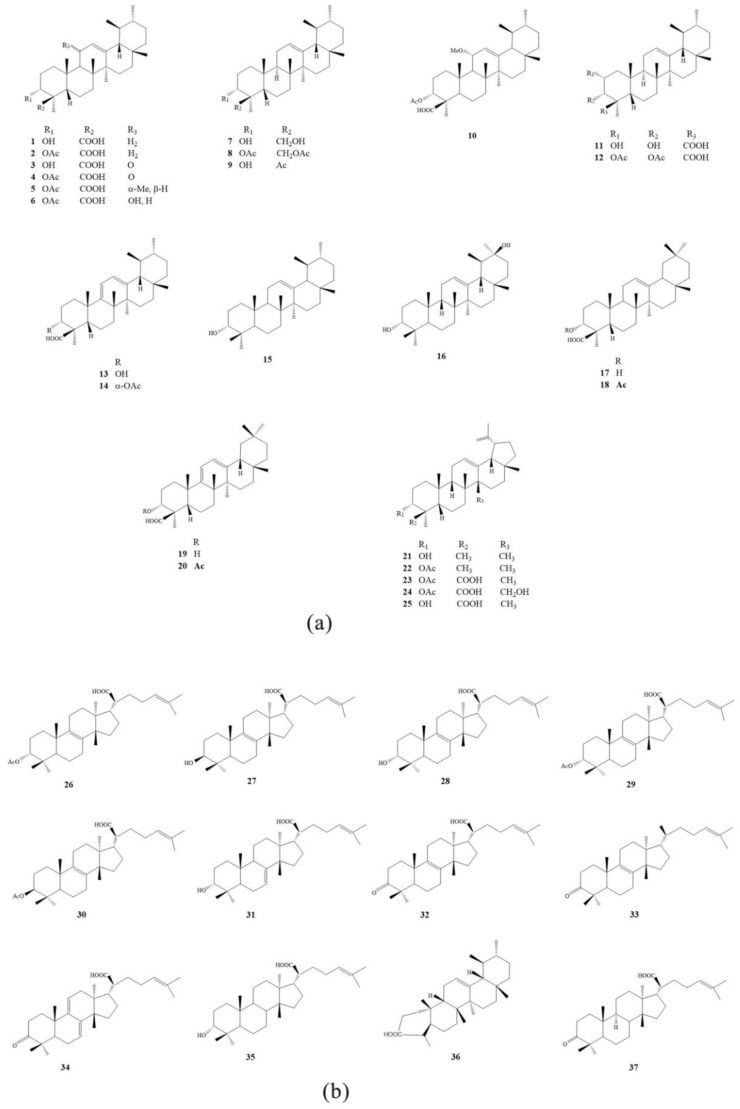
The structural patterns of some chemical constituent in frankincense. (**a**) Pentacyclic triterpenoids; (**b**) tetracyclic triterpenoids; (**c**) macrocyclic diterpenoids; (**d**) essential oils.

**Figure 2 molecules-24-03076-f002:**
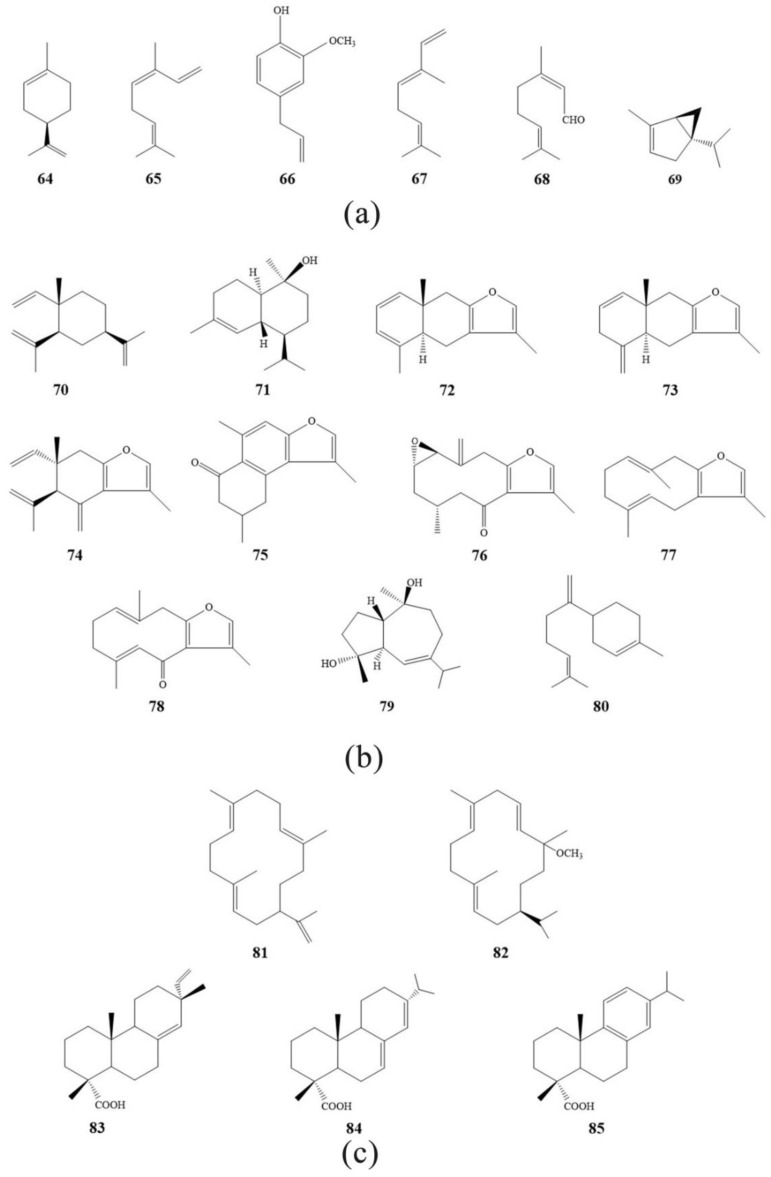
The structural patterns of some chemical constituents in myrrh. (**a**) Monoterpenes; (**b**) sesquiterpenes; (**c**) diterpenoids; (**d**) triterpenoids; (**e**) steroids; (**f**) lignans.

**Figure 3 molecules-24-03076-f003:**
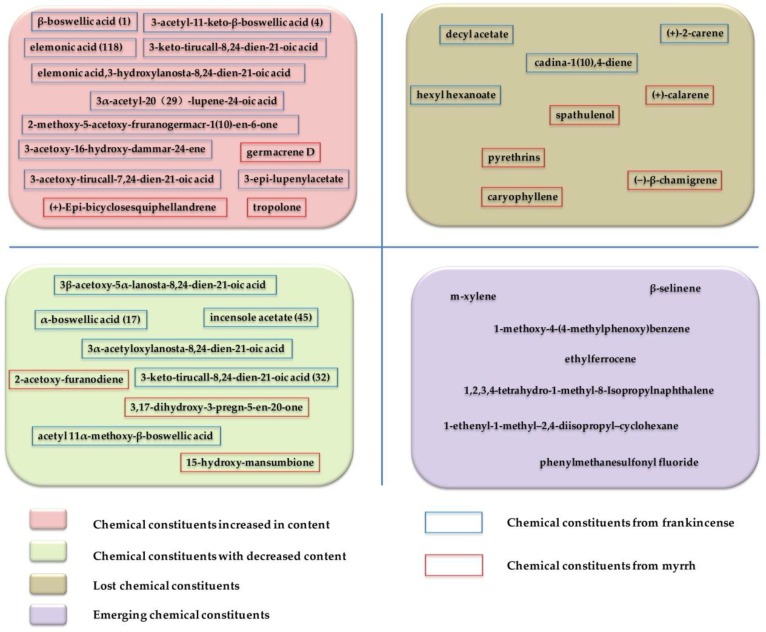
Changes in chemical constituents of frankincense and myrrh before and after compatibility.

**Figure 4 molecules-24-03076-f004:**
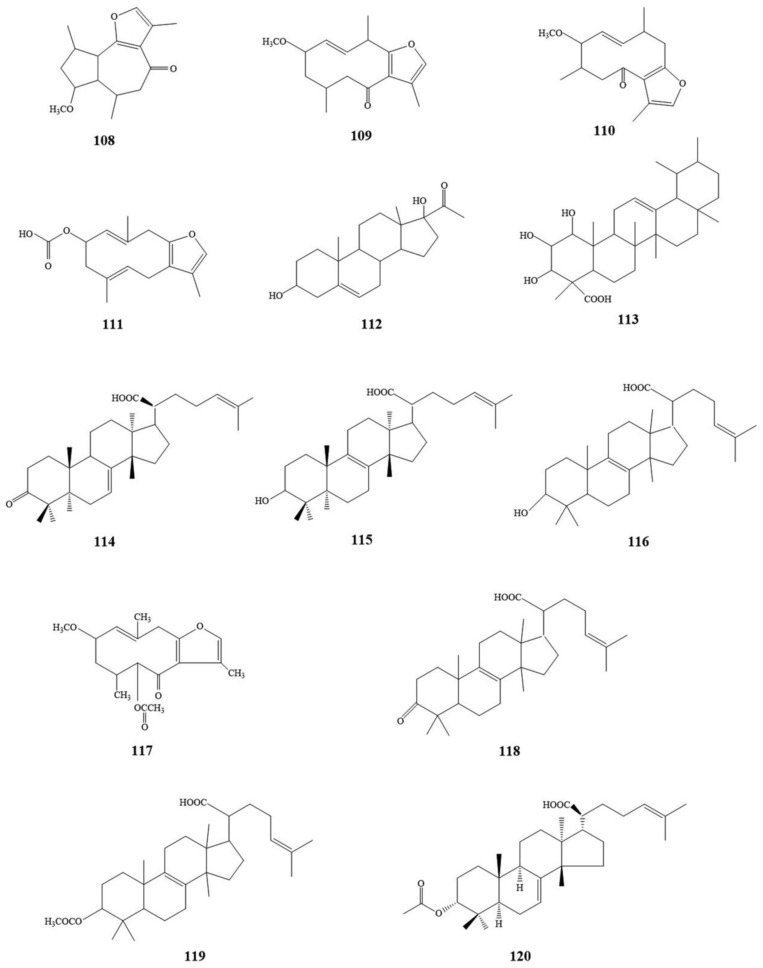
The chemical structure of some potential bioactive components in frankincense–myrrh compound.

**Figure 5 molecules-24-03076-f005:**
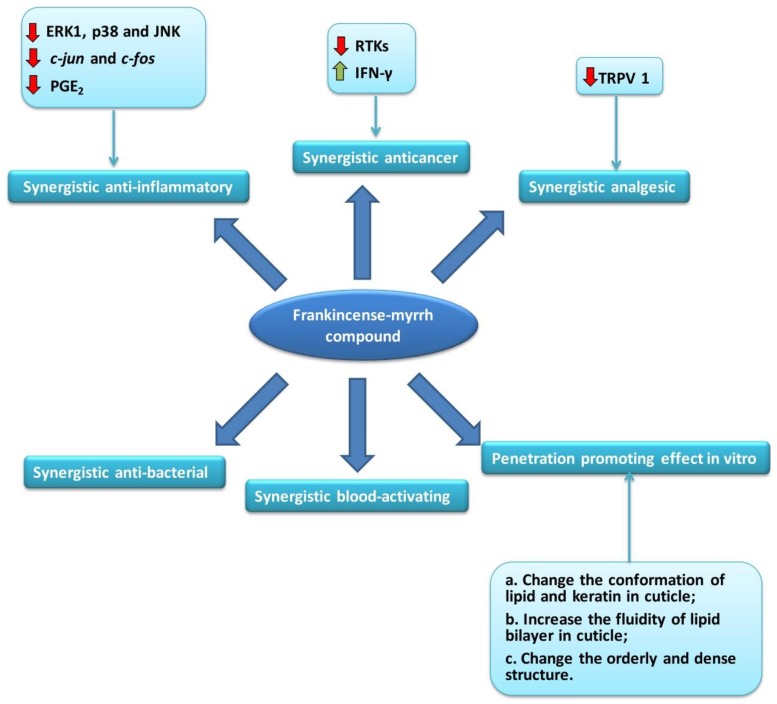
The pharmacological effects and mechanisms of frankincense–myrrh compound.

**Figure 6 molecules-24-03076-f006:**
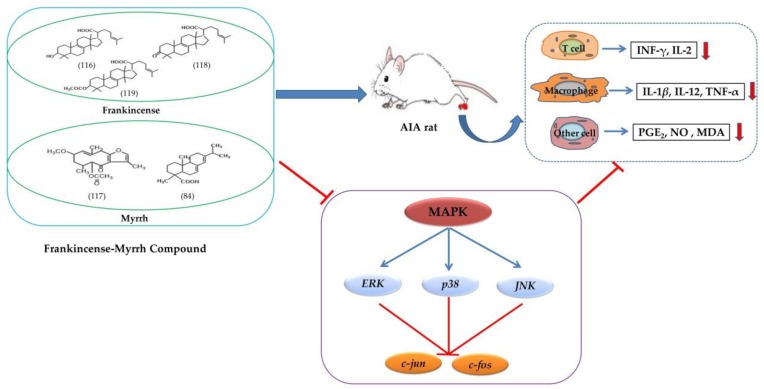
Illustration of frankincense–myrrh compound in the treatment of adjuvant-induced arthritis (AIA) in rats. The bioactive components of frankincense and myrrh significantly reduced the expression levels of inflammatory cytokines INF-γ, IL-2, IL-1β, IL-12, TNF-α, PGE_2_, NO, and MDA in AIA rats’ serum and foot swelling by reducing the phosphorylation of three kinds of MAPK (ERK, p38, and JNK) and the down-regulation of downstream genes (*c-jun* and *c-fos*).

**Figure 7 molecules-24-03076-f007:**
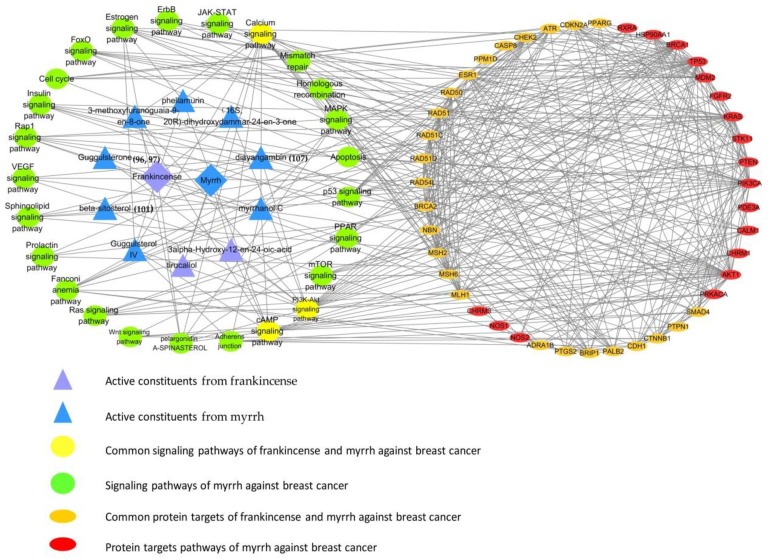
Active constituent-protein targets-signaling pathway network of frankincense–myrrh compound against breast cancer.

**Table 1 molecules-24-03076-t001:** Main chemical constituents and plant sources of frankincense.

No.	Chemical Class	Chemical Name	Plant Sources	Ref.
1	Pentacyclic triterpenes	β-boswellic acid	*Boswellia carterii; Boswellia serrata*	[22]
2		acetyl-β-boswellic acid	*Boswellia carterii; Boswellia serrata*	[23]
3		11-keto-β-boswellic acid	*Boswellia carterii; Boswellia serrata*	[24]
4		3-acetyl-11-keto-β-boswellic acid	*Boswellia carterii; Boswellia serrata*	[25]
5		3α-acetyl-11-methoxy-12-ursane ene-24-acid	*Boswellia serrata*	[26]
6		acetyl-11- hydroxyl-β-boswellic acid	*Boswellia carterii*	[27]
7		urs-12-ene-3,23-diol	*Boswellia serrata*	[28]
8		urs-12-ene-3,23-diol, diacetate	*Boswellia carterii; Boswellia serrata*	[28]
9		3β-hydroxy-24-norurs-12-en-4-ylmethyl	*Boswellia carterii; Boswellia serrata*	[29]
10		acetyl-11-α-methoxy-β-boswellic acid	*Boswellia serrata*	[26]
11		2α,3α-dihydroxyurs-12-ene-24-oic acid	*Boswellia serrata*	[28]
12		urs-12-ene-23-oic acid	*Boswellia. serrata*	[28]
13		9,11-dehydro-β-boswellic acid	*Boswellia serrata*	[30]
14		3α-acetyl-9,11-dehydro-β-boswellic acid	*Boswellia carterii; Boswellia serrata*	[31]
15		α-amyrin	*Boswellia carterii; Boswellia serrata*	[32]
16		3α,20β,18Hβ-ursane diol	*Boswellia serrata*	[33]
17		α-boswellic acid	*Boswellia carterii; Boswellia serrata*	[34]
18		acetyl-α-boswellic acid	*Boswellia carterii; Boswellia serrata*	[27]
19		9, 11-dehydro-α-boswellic acid	*Boswellia carterii; Boswellia serrata*	[35]
20		3-acetyl-9,11-dehydro-α-boswellic acid	*Boswellia serrata*	[35]
21		lupeol	*Boswellia carterii*	[36]
22		lupeolacetate	*Boswellia carterii*	[36]
23		acetyl lupeolic acid	*Boswellia carterii*	[30]
24		epilupeol	*Boswellia serrata*	[35]
25		lupeolic acid	*Boswellia carterii*	[36]
26	Tetracyclic triterpenoids	acetyl-α-elemolic acid	*Boswellia serrata*	[37]
27		3β-hydroxy-tirucalla-8,24-dien-21-oic acid	*Boswellia carterii; Boswellia serrata*	[30]
28		3α-hydroxy-tirucalla-8,24-dien-21-oic acid	*Boswellia carterii; Boswellia serrata*	[30]
29		elemolic acid	*Boswellia carterii; Boswellia serrata*	[37]
30		3-oxo-tirucallic acid	*Boswellia carterii; Boswellia serrata*	[38]
31		3α-hydroxy-tirucalla-7,24-dien-21-oic acid	*Boswellia carterii; Boswellia serrata*	[39]
32		3-keto-tirucalla-8,24-dien-21-oic acid	*Boswellia carterii; Boswellia serrata*	[37]
33		kanziol	*Boswellia carterii*	[36]
34		3-methoxy-tirucalla-7,9(11),24-trien-21-oic acid	*Boswellia carterii; Boswellia serrata*	[33]
35		3α-hydroxy-tirucalla-24-ene-21-oic acid	*Boswellia carterii; Boswellia. serrata*	[33]
36		3,4-secours-12-en-3-oic acid	*Boswellia carterii*	[40]
37		3α-keto-tirucalla-24-ene-21-oic acid	*-*	[41]
38	Macrocyclic diterpenoids	cembrene	*Boswellia carterii*	[42]
39		cembrene C	*Boswellia carterii; Boswellia serrata*	[43]
40		cembrene A	*Boswellia carterii; Boswellia serrata*	[43]
41		serratol	*Boswellia serrata*	[37]
42		sarcophytol	*Boswellia carterii; Boswellia serrata*	[37]
43		3,7,11-cyclotetradecatrien-1-ol-4,8,12-trimethyl-1-(1-methylethyl)-acetate	*Boswellia carterii; Boswellia serrata*	[37]
44		incensole	*Boswellia carterii; Boswellia serrata*	[42]
45		incensole acetate	*Boswellia carterii; Boswellia serrata*	[44]
46		lncensole-oxide	*Boswellia. carterii; Boswellia serrata*	[45]
47		acetyl incensole-oxide	*Boswellia carterii; Boswellia serrata*	[43]
48		isoincensolol	*Boswellia carterii; Boswellia serrata*	[43]
49		isoincensolol acetate	*Boswellia carterii; Boswellia serrata*	[46]
50		isoincensolol-oxide	*Boswellia carterii; Boswellia serrata*	[45]
51		verticilla-4(20),7,11-triene	*Boswellia carterii; Boswellia serrata*	[43]
52	Essential oils	α-pinene	*Boswellia carterii; Boswellia serrata*	[47]
53		β-pinene	*Boswellia carterii; Boswellia serrata*	[42]
54		camphene	*Boswellia carterii; Boswellia serrata*	[42]
55		α-thujene	*Boswellia carterii; Boswellia serrata*	[42]
56		sabinene	*Boswellia carterii; Boswellia serrata*	[42]
57		myrcene	*Boswellia carterii; Boswellia serrata*	[43]
58		p-cymene	*Boswellia carterii; Boswellia serrata*	[43]
59		limonene	*Boswellia carterii; Boswellia serrata*	[43]
60		α-phellandrene	*Boswellia carterii; Boswellia serrata*	[43]
61		linalool	*Boswellia carterii; Boswellia serrata*	[43]
62		octyl acetate	*Boswellia carterii; Boswellia serrata*	[48]
63		1-octanol	*Boswellia carterii; Boswellia serrata*	[49]

**Table 2 molecules-24-03076-t002:** The pharmacological effects and mechanisms of frankincense.

Pharmacological Effects	Extracts or Compounds	Mechanism
Anti-inflammatory	KBA (**3**)	↓ 5-LO, ↓ HLE
	Extract of frankincense	↓ 5-LOX
	AKBA (**3**) and Arsenic trioxide	↓ MMP-1, MMP-2, MMP-9 ; ↓ TNF-α; ↓ IL-β
	Extract of frankincense	↓ Proinflammatory cytokines
	Bas	↓ Antimicrobial peptide LL-37
	Frankincense oil extract	↓ COX-2; Inhibiting inflammatory infiltration induced by noxious stimulation
Anticancer	Frankincense oil	Specifically blocking the growth cycle of J82 cells
	Frankincense oil	↑ bax/bcl-2
	Frankincense oil	Regulating of AMPK/mTOR pathway
	incensole acetate (**45**); incensole (**44**)	—
	incensole acetate (**45**)	—
	AKBA (**4**)	↓ NF-κB
	AKBA (**4**)	↓ VEGFR2
	AKBA (**4**)	—
	AKBA (**4**)	↑ DR5
	AKBA (**4**)	↓ CXCR4
	AKBA (**4**)	Demethylation and reactivation of methylation silenced tumor suppressor genes
	KBA (**3**)	—
	KBA (**3**)	—
	KBA (**3**)	↑ let-7, ↑ miR-200microRNA
Antiulcer	AKBA (**4**)	↓ MMPs
Improving memory	Frankincense oil extract	—
Antioxidant	Bas	Regulating the Nrf2/HO-1 pathway

Abbreviations: 5-LO: 5-Lipoxygenase; HLE: Human leukocyte elastase; 5-LOX: 5-Liperoxidase; MMP-1: Matrix metalloproteinase-1; MMP-2: Matrix metalloproteinase-2; MMP-9: Matrix metalloproteinase-9; TNF-α: Tumor necrosis factor-α; (IL-β): Interleukin-1beta; COX-2: Cyclooxygenase-2; bax: BCL2-Associated X Protein; bcl-2: B-cell lymphoma-2; AMPK: Adenosine 5‘-monophosphate (AMP)-activated protein kinase; mTOR: Mechanistic target of rapamycin; NF- kB: Nuclear factor-kappa B; VEGFR2: Vascular endothelial growth factor receptor 2; DR5: Death receptor 5; CXCR4: Chemokine receptor 4; Nrf-2: The nuclear factor erythroid 2 (NFE2)-related factor 2; HO-1: Heme oxygenase-1; KBA: 11-keto-β-boswellic acid; AKBA: 3-acetyl-11-keto-β-boswellic acid.

**Table 3 molecules-24-03076-t003:** The main chemical constituents and plant sources of myrrh.

No.	Chemical Class	Chemical Name	Plant Sources	Ref.
64	Monoterpenes	limonene	*Commiphora quadricincta*	[71]
65		cis-β-ocimene	*Commiphora quadricincta*	[71]
66		eugenol	*Commiphora mukul*	[72]
67		trans-β-ocimene	*Commiphora tenuis*	[73]
68		(2Z)-3,7-dimethyl-2,6-octadienal	*Commiphora tenuis*	[73]
69		α-thujone	*Commiphora tenuis*	[73]
70	Sesquiterpenes	β-elemene	*Commiphora myrrha*	[74]
71		T-cadinol	*Commiphora guidottii*	[75]
72		furanocudesma-1,3-diene	*Commiphora myrrha*	[76]
73		lindestrene	*Commiphora myrrha*	[76]
74		curzerene	*Commiphora myrrha*	[77]
75		myrrhone	*Commiphora myrrha; Commiphora opobalsamum*	[78]
76		rel-1S,2S-epoxy-4R-furanogermacr-10(15)-en-6-one	*Commiphora myrrha*	[79]
77		furanodiene	*Commiphora myrrha*	[80]
78		1(10), 4-furanodien-6-one	*Commiphora erythraea*	[81]
79		guaianediol	*Commiphora opobalsamum*	[82]
80		β-bisabolene	*Commiphora guidottii*	[83]
81	Diterpenoids	(dl)-cembrene A	*Commiphora mukul*	[84]
82		(1E,4E,8E)-4,8,14-trimethyl-11-(1-methylethyl)-14-methoxycyclotetradeca-1,4,8-triene	*Commiphora mukul*	[85]
83		sandaracopimaric acid	*Commiphora myrrha*	[86]
84		abietic acid	*Commiphora myrrha*	[86]
85		dehydroabietic acid	*Commiphora myrrha*	[86]
86	Triterpenoids	β-amyrin	*Commiphora kua*; *Commiphora confuse*	[87]
87		cycloartan-24-ene-1α,2α,3β-triol	*Commiphora opobalsamum*	[82]
88		3-epi-α-amirone	*Commiphora myrrha*	[76]
89		mansumbinone	*Commiphora myrrha*	[76]
90		3,4-seco-mansumbinoic acid	*Commiphora molmol*	[88]
91		mansumbinoic acid	*Commiphora* *kua*	[89]
92		myrrhanone A	*Commiphora mukul*	[90]
93		myrrhanone B	*Commiphora mukul*	[91]
94		3β-hydroxydammar-24-ene	*Commiphora confuse*	[92]
95		3β-acetoxycycloartan-24-ene-1α,2α-diol	*Commiphora opobalsamum*	[93]
96	Steroids	(Z)-guggulsterone	*Commiphora mukul*	[94]
97		(E)-guggulsterone	*Commiphora mukul*	[94]
98		guggulsterone-M	*Commiphora wightii*	[95]
99		guggulsterol-Y	*Commiphora wightii*	[95]
100		pregn-4-ene-3,16-dione	*Commiphora mukul*	[85]
101		β-sitosterol	*Commiphora sphaerocarpa*	[93]
102	Lignans	erlangerins A	*Commiphora erlangeriana*	[96]
103		erlangerins B	*Commiphora erlangeriana*	[96]
104		(+)-sesamin	*Commiphora mukul*	[84]
105		picropolygamain	*Commiphora incisa*	[97]
106		polygamain	*Commiphora incisa*	[97]
107		diayangambin	*Commiphora mukul*	[90]

**Table 4 molecules-24-03076-t004:** Pharmacological effects and mechanisms of myrrh.

Pharmacological Effects	Extracts or Compounds	Mechanisms
Anti-inflammatory	guggulsterone (**96**, **97**)	↓ ERK, ↓ JNK
	mansumbinoic acid (**91**)	—
	1(10), 4-furanodien-6-one (**78**)	↓ NF-κB
Anticancer	β-elemene (**70**)	↑ p38MAPK
	rel-1S,2S-epoxy-4R-furanogermacr-10(15)-en-6-one (**76**)	—
	β-bisabolene (**80**)	—
	cycloartan-24-ene-1α,2α,3β-triol (**95**)	—
	Extract of myrrh	↑ Nrf2/HO-1; Reducing inflammation, angiogenesis, and oxidative stress
	guggulsterone (**96**, **97**)	↓ IAP1, xIAP, Bfl-1/A1, Bcl-2, cFLIP and survivin
	guggulsterone (**96**, **97**)	Activation of intrinsic mitochondria pathway
Analgesic effect	furanocudesma-1,3-diene (**72**); curzerene (**74**)	Action on opioid receptors in the brain
	Extract of myrrh	—
	Extract of myrrh	—
Antibacterial	Myrrh essential oil	—
	3,4-seco-man-sumbinoic acid (**90**)	—

Abbreviations: ERK: Extracellular regulated protein kinases; JNK: c-Jun N-terminal kinase; NF- kB: nuclear factor-kappa B; MAPK: Mitogen-activated protein kinase; Nrf-2: The nuclear factor erythroid 2 (NFE2)-related factor 2; HO-1: Heme oxygenase-1; IAP1: Inhibitor of apoptosis protein-1; xIAP: x-linked inhibitor of apoptosis protein; Bfl-1/A1: Bcl-2-related protein A1; Bcl-2: B-cell lymphoma-2.

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
