# Peer review of "Seeing the Unseen of the Combination of Two Natural Resins, Frankincense and Myrrh: Changes in Chemical Constituents and Pharmacological Activities"

_molecules, 2019, doi:10.3390/molecules24173076_

Round 1

Reviewer 1 Report

This review submitted by Cao et al., entitled “Seeing the Unseen of the Combination of Two Natural Resin Frankincense and Myrrh: Changes in Chemical Constituents and Pharmacological Activities” deals about the different therapeutic effects of frankincense and myrrh, two natural resins as well as their synergistic effect for different therapeutic potentials. The discovery and characterization of new drugs especially from natural sources is worthy of investigation because of the emergence of new diseases, resistance to current therapies and side effect of drugs. The manuscript is clear and well written. However, I think that at the present time the manuscript needs corrections, modifications and additional information.

Editing and comments

Line 33:

“…the disappearance of old chemical components…”

The term “old” is not appropriate in this situation, I would therefore recommend the term “native”, “starting material” “original secondary metabolites”,…

Line 57:

Please replace “Boswellia Carterii” with “Boswellia carterii”.

Line 58:

You mention that myrrh is a substance exuded from the bark of Commiphora plant. Please mention the Latin name of this plant as you did for frankincense.

Line 105:

In the table you mention B. serrata as source of frankincense. Is there any reason why this name appear here and there is no mention in the introduction? Is frankincense secreted by only one tree or several species? Please correct.

Line 118:

What “5-LO” is abbreviated for?

Figure 1?

Is there any Figure 1? If not please correct the numbering.

Line 127:

Please replace “proinflammatory” with “pro-inflammatory”.

Line 129, 131 and manuscript

Please choose one construction “antimicrobial” (line 129) or anti-microbial (132) and keep using it. Please check you manuscript for similar terms “anticancer”, “anti-cancer”, “antiinflammatory”, “antibacterial”,…

Line 138

Please write in whole term “COX-2” and feel free after to use the abbreviation in your manuscript.

Line 145:

Please confirm “…the growth and apoptosis of J82 cells can be inhibited.”

Line 190 (Table 2)

Please include the numeric code for the different compounds when it is possible.

E.g. AKBA (4).

Please also consider this remark for the manuscript each time you mention a compound present in your figure 2 and 3.

Please include and detail abbreviation for the different terms (5-LO, HLE,5-LOX,…) in the legend.

Figure 3

Please include a classification (a, b, c,…) as you did for figure 2.

Please include legend at bottom of the figure.

Paragraph 3.2.1

Please replace “Guggulsterone” with “guggulsterone”.

Paragraph 3.2.2

Please confirm “PC3 and DU-145 human breast cancer cells”.

What do you mean by “it induces apoptosis by activating mitochondria”. Please explain.

Paragraph 3.2.4

Please use italic for the name of the different microbial organisms.

Table 4:

Same comment as table 2 (line 190).

Please include the numeric code for the different compounds when it is possible.

Please include and detail abbreviation for the different terms (ERK, IAP1,…) in the legend.

Paragraph 4.2.6

What is the meaning of “Chuanxiong”?

Figure 4:

Please include code to molecular structure on figure 2, 3 and 5.

Please find a way to mention or highlight (color code/ shape?) which compounds are from frankincense or from myrrh.

If the mechanism of formation of new compounds is known or if the starting material are known, please mention them (table?).

Paragraph 4.1.1

Please include the numeric code for the different compounds when it is possible.

Paragraph 4.1.3 (1st line)

What is the meaning of “CWE”? Please explain.

Figure 7:

Are the 5 different compounds from frankincense or from myrrh or are they products of the reaction? Please also include the numerical code for each compound.

Please increase the font size of your figure.

What is the meaning of “MWE” and “FMW”?

Figure 8:

Please reshape your figure in order that the text does not overlap other text and shape.

Can you please adapt your figure in order to identify (color code?) which pathways are up-regulated or down-regulated by frankincense - myrrh compounds?

Author Response

Thank you very much for the comments, please find the attached response.

Reviewer 2 Report

This manuscript review the chemical constitutes and bio-activities of two resin. The synergistic effect may be important and potential to be researched. My suggestions are below:

where is the figure 1.? In the whole manuscript, the number of compound should be bold. In Table 1 and 3, I suggest that full chemical name should been written, rather than representative compounds. In figure 2. Compound 5, 9, 11, 14, and 23 should not written alpha or beta, since the structures are already shown. Compound 27-29, the structure is beta, write alpha or beta would be confused. line 117, 118, 123, and 129, the KBA, 5-LO… the full name should been written before abbreviation for the first time in the whole manuscript. Figure 3. Compound 105 and 106, where is the bold bond and dash bond?

Author Response

(The authors gave the same response as above.)
